# DENSE VIDEO UNDERSTANDING WITH GATED RESIDUAL TOKENIZATION

## ABSTRACT

High temporal resolution is essential for capturing fine-grained details in video understanding. However, current video large language models (VLLMs) and evaluation benchmarks predominantly rely on low-frame-rate sampling, such as uniform sampling or frame selection, which discards dense temporal information. This compromise is primarily made to avoid the high computational cost of tokenizing every frame, which leads to redundant computation during frame-level tokenization and a linear increase in token count as video length grows. Such a trade-off stems from engineering constraints in existing video understanding systems that rely on frame selection and sampling. Yet, for tasks such as lecture or educational video comprehension, where information is distributed across nearly every frame, this compromise becomes a major limitation. These tasks require frame-by-frame reasoning and fine-grained temporal alignment, and current approaches discourage progress on high-frame-rate datasets or models. To address this gap, we introduce the novel task of Dense Video Understanding, which aims to enable video comprehension at high frame rates. Our goal is to reduce the tokenization time of high-FPS videos and minimize the token overhead incurred by dense frame sampling. This lack of dense modeling also affects current benchmarks, whose question-answer pairs are often designed around slowly changing content, making them insufficient for evaluating fine-grained temporal understanding. To this end, we propose the first benchmark specifically tailored for dense video understanding: DIVE (Dense Information Video Evaluation). To overcome inefficiencies in frame-wise tokenization, we propose Gated Residual Tokenization (GRT), a two-stage token acceleration and reduction framework that operates both during and after tokenization, addressing inefficiencies at the inter-tokenization and intra-tokenization levels, respectively: First, Motion-Compensated Inter-Gated Tokenization applies pixel-level motion estimation and a gating mechanism during tokenization to identify and skip static regions, encoding only the moving patches. This results in sub-linear growth in both tokenization time and token count. Second, Semantic-Scene Intra-Tokenization Merging performs content-level token merging across static regions within a scene, further reducing redundancy while preserving dynamic semantic content. Extensive experiments on the DIVE benchmark show that our methods not only outperform larger VLLM baselines but also consistently improve as FPS increases. These results underscore the importance of preserving dense temporal information and demonstrate that GRT enables scalable, efficient high-FPS video understanding.

## 1 INTRODUCTION

Does visual frame density matter for humans or for video large language models? The short answer is yes. Visual frame density, the number of frames captured per second, has long been essential for both biological and artificial vision systems. Many species rely on high-frame-rate vision to detect rapid changes to survive in their environment, and engineers invest heavily in high-speed camera (Litzenberger et al., 2007) technology to capture fine-grained motion (Felsen et al., 2018). Audiences likewise prefer high-frame-rate video and gaming content for smoother, more immersive experiences. Together, these observations underscore the importance of dense temporal sampling for accurate perception and information extraction.

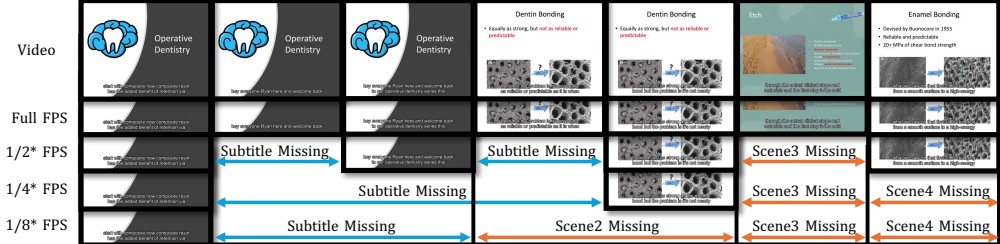

Figure 1: Visualization on impact of a frame-rate reduction on video question answering. Most current models sample videos with a strict frame limit and at fixed time intervals, resulting in a very low effective frame rate. Here, we use subtitle reading (OCR (Fei et al., 2025)) in an educational video benchmark to illustrate the impact of frame rate reduction. Frame-by-frame analysis is required to capture all subtitles; when the frame rate is halved, some subtitles are missed and short scene segments become invisible to the video large language model. As the frame rate decreases further, more scenes are skipped. At one-eighth of the original rate, only one frame remains per segment, rendering frame-by-frame reasoning impossible.

In video understanding research, large language models (LLMs) must interpret dynamic visual streams by converting frames into token sequences. However, most existing approaches drastically reduce the input frame rate, sampling only a handful of frames per second, and discard the majority of temporal information. As illustrated in Fig. 1, this coarse sampling can cause critical details, such as OCR-detected subtitles or brief instructional segments to vanish. At extreme subsampling rates (one-eighth of the original), each segment may contain only a single frame: sufficient to convey a general scene, but wholly inadequate for tasks that require frame-by-frame reasoning.

Nonetheless, existing video large language models (vLLMs) (Li et al., 2024; Zhang et al., 2024b; 2023; Maaz et al., 2024) typically sample only a sparse subset of frames at low frames per second (FPS) before processing, thereby discarding the dense temporal information contained in the omitted frames. Many of these methods enforce a maximum frame budget, which effectively reduces FPS as video duration increases. Remarkably, some approaches even report state-of-the-art results using as few as six frames per clip (Kim et al., 2024). These findings suggest that current video understanding benchmarks do not demand high temporal resolution to achieve strong performance—an observation that directly contradicts our analysis in Fig. 1 and challenges the broader motivation for fine-grained, high-FPS video understanding. This mismatch underscores the need for a new task formulation: one that explicitly focuses on dense temporal video understanding. We argue for the development of Dense Video Understanding as a necessary step toward evaluating and building models that can effectively process and reason over high-frame-rate video content.

We attribute this oversight to three fundamental challenges in current vLLMs research. **First**, high-frame-rate videos produce an excessive number of tokens after standard patch-based tokenization, quickly overwhelming the limited throughput of existing LLMs (Zhu et al., 2024). Since the self-attention mechanism scales quadratically (Zhang & Fu, 2025) with token count, models rapidly become intractable as FPS increases. Without an effective token reduction strategy, no current video-LLM can efficiently process full-FPS inputs. **Second**, popular tokenizers such as CLIP (Radford et al., 2021) and SigLIP (Zhai et al., 2023), which rely on convolutional layers, are designed to process entire images. Each image must be passed through the full convolutional layers before being split into patch tokens. This design hinders parallelism across dense video frames and results in linear growth in both tokenization time and token count as FPS increases. Such an approach limits scalability and prevents models from accessing raw visual signals efficiently. What is needed instead is a gated inter-tokenizer that filters out uninformative patches before embedding—achieving sub-linear cost by selecting only the most informative regions. **Third**, prevailing video understanding benchmarks are tailored to the limitations of current models. They primarily pose coarse-grained queries, such as activity recognition or object presence, that can be answered using just a handful of frames. For example, recent work (Kim et al., 2024) reports state-of-the-art performance with only six frames per clip. This simplicity masks the need for dense temporal reasoning and fails to evaluate models' ability to process high-FPS content. To bridge this gap, a new benchmark specifically designed for dense video understanding is urgently needed.

To address these challenges, we make three key contributions. First, we propose the novel task of **Dense Video Understanding**, which aims to equip models with the capability to comprehend densely sampled, high-FPS video content. To address the lack of appropriate benchmarks for this setting, we introduce **DIVE** (Dense Information Video Evaluation), the first benchmark explicitly designed for high-FPS video question answering. DIVE consists of densely sampled clips paired with QA tasks that require frame-by-frame reasoning—where skipping even a few frames leads to immediate information loss (see Fig.1). Details on benchmark build details are provided in Sec.4.

Second, we present **Gated Residual Tokenization** (GRT), a two-stage token acceleration and reduction framework. It begins with Motion-Compensated Gated Inter-Tokenization, which removes convolutional layers in tokenizers like CLIP and SigLIP to allow for patch-wise, parallel tokenization. Using per-patch motion masks, our method filters out static regions before tokenization, embedding only dynamic patches with a pretrained ViT-based tokenizer. This yields sub-linear tokenization cost as FPS increases (Sec. 3.2).

Third, we propose a Semantic-Scene Token Merging module that further compresses token sequences at the semantic level, complementing the pixel-level redundancy removal performed by Inter-Tokenization. After extracting key-frame and P-frame token sets, we compute distributional similarity across frames to merge semantically redundant key tokens, while preserving motion-specific P-frame tokens. This reduces sequence length without sacrificing critical spatiotemporal information (Sec. 3.3).

Our main contributions are summarized as follows:

- **Dense Video Understanding Task:** We propose the first task specifically designed to evaluate video understanding on high-FPS content, addressing the limitations of prior work that relied on uniform sampling and sparse frame selection while neglecting dense temporal information.

- **DIVE Benchmark:** We introduce **DIVE** (Dense Information Video Evaluation), the first benchmark for high-FPS video question answering. DIVE features densely sampled video clips and QA pairs requiring true frame-by-frame reasoning, bridging the gap left by coarse-resolution benchmarks.

- **Gated Residual Tokenization (GRT):** We present a two-stage framework for accelerating and reducing tokenization in dense video settings: 1. **Motion-Compensated Gated Inter-Tokenization** filters out uninformative patches before tokenization using per-pixel motion masks. To enable patch-level parallelization, we replace convolutional layers in conventional tokenizers with lightweight pretrained MLPs. This design prevents redundant tokens from entering the model and achieves sub-linear complexity with respect to FPS. 2. **Semantic-Scene Token Merging** further compresses token sequences by clustering semantically similar key-frame tokens based on distributional similarity, while preserving motion-related P-frame tokens. This preserves essential spatiotemporal information while reducing sequence length.

- **Empirical Validation:** Our 0.5B-parameter model achieves state-of-the-art performance on DIVE, with MOS scores consistently improving as FPS increases. This demonstrates the value of dense temporal cues and the scalability of our gated tokenization and merging framework for video-LLMs.

## 2 Related Work

### 2.1 Frame Sampling in Video Large Language Models

Video large language models (vLLMs) extend vision–language pretraining to dynamic inputs, enabling tasks such as video question answering and captioning. Early works like Flamingo (Alayrac et al., 2022) and MERLOT Reserve (Zellers et al., 2022) demonstrated the feasibility of aligning video frames with text, while recent architectures such as LLaVA and its variants (Li et al., 2024; Liu et al., 2024) adapt large language models (LLMs) to video by inserting vision embeddings into transformer layers. Despite these advances, most vLLMs rely on sparse temporal sampling—selecting a fixed number of frames per clip. For example, LLaVA-One Vision limits clips to 32 frames, LLaVA-Vid to 20 frames, and VideoLLaVA to just 8 frames (Kim et al., 2024). MovieChat (Song et al., 2024) and MovieChat+ (Song et al., 2025) first extend to over ten thousands frame. Such caps simplify computation but discard dense temporal cues, preventing models from performing frame-by-frame reasoning on high–FPS content.

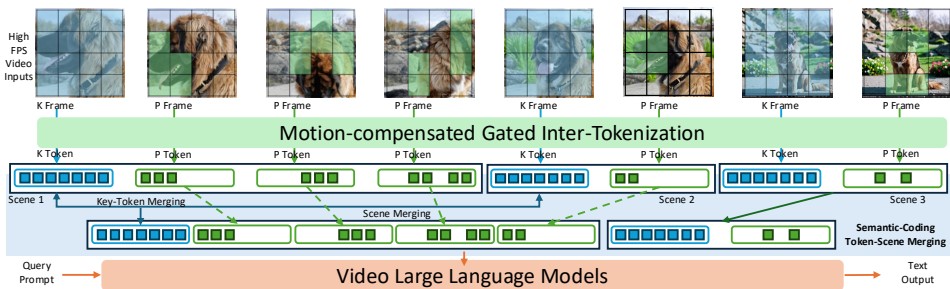

Figure 2: Overall architecture of our gated residual tokenization pipeline. Given an input video, we first apply a pixel-coding, motion-compensated gated tokenization process: key frames extract static scene tokens, while P-frames capture moving patches as P-token sets. Each scene thus yields one key-token set and multiple P-token sets. A subsequent semantic-coding token-scene merging module measures similarity between adjacent scenes (via their key-token distributions) and merges ones that are semantically equivalent by clustering key tokens into a new representative token and concatenating the P tokens. The resulting reduced token sequence, together with the query prompt tokens, is then fed into a video large-language model to generate the final answer.

## 2.2 PATCH-BASED TOKENIZATION AND FRAME SELECTION

Standard vLLM pipelines tokenize each selected frame into a grid of patch embeddings (e.g., 16×16 patches in ViT) before feeding them into the language model. Frame selection is typically based on uniform sampling or simple motion heuristics (Xue et al., 2022; Wang et al., 2023), but these strategies still process every patch in each sampled frame—yielding long token sequences and high preprocessing latency. AuroraCap (Chai et al., 2024) and AuroraLong (Xu et al., 2025) conduct patch-level merging for better efficiency. Adaptive sampling methods (Kim et al., 2024) allocate more frames to high-motion segments, yet they do not address the quadratic self-attention cost arising from large token budgets. Moreover, even flexible-FPS variants retain caps on total frames, limiting their ability to leverage very high frame rates.

## 3 METHODOLOGY

### 3.1 NEW TASK DEFINITION: DENSE VIDEO UNDERSTANDING

High-FPS video content is crucial for capturing fine-grained temporal dynamics. However, it has been largely overlooked in video understanding research and in video-LLM design, due to the lack of suitable benchmarks and the prohibitive length of resulting token sequences. We define *Dense Video Understanding* as the task of processing all frames in a high-FPS video without imposing a maximum frame limit. Unlike conventional sparse-sampling strategies, this task preserves the full temporal resolution of the input.

Formally, given a high-FPS video $v$, a selector retains all frames and a tokenizer converts them into a sequence of visual tokens:

$$\tau(v) = \text{Tokenize}\big(\text{SelectHighFPS}(v)\big) \tag{1}$$

These visual tokens, together with the text tokens $T$ representing the query, are then fed into a video LLM to generate the answer:

$$\hat{y} = \text{LLM}\big(\tau(v), T\big) \tag{2}$$

This formulation ensures that all temporal information is preserved and available to the model.

### 3.2 MOTION-COMPENSATED GATED INTER-TOKENIZATION

#### 3.2.1 PIXEL-CODING VIDEO SCENE REPRESENTATION

Existing video LLMs tokenize every frame and then prune or merge redundant tokens, resulting in unnecessary computation. To exploit the strong temporal redundancy in high-FPS videos, we draw

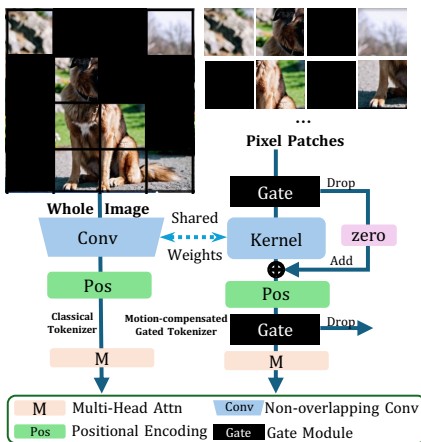

**Pixel Patches**

Figure 3: Comparison of classical tokenizers, such as CLIP (Radford et al., 2021) and SigLIP (Zhai et al., 2023), with our Motion-Compensated Gated Inter-Tokenizer (MCG Tokenizer). **Left:** classical tokenizer applying convolutions over the entire image—including masked patches—leading to high computational cost. **Right:** our MCG Tokenizer reuses pretrained kernels as MLPs, applies gating to filter irrelevant K- and P-patches (zero-tensor placeholders for positional encoding), and feeds only valid tokens into attention, greatly reducing tokenization time.

inspiration from classical video compression techniques, which encode scenes using one full *key frame* and a series of *P-frames* that capture only the changing patches. Let $f_{s,k}$ denote the key frame of scene $s$, and let $f_{s,k+i}$ be the $(i+1)$-th frame. We define the P-frame residual as:

$$\Delta f_{s,k+j} = M_{s,k+j} \odot \left( f_{s,k+j} - f_{s,k+j-1} \right), \quad (3)$$

where $M_{s,k+j} \in \{0,1\}^{H \times W \times C}$ is a binary mask that selects only the pixel patches that have changed from the previous frame. Using these residuals, any frame in the scene can be reconstructed nearly losslessly:

$$f_{s,k+i} = f_{s,k} + \sum_{j=1}^{i} \Delta f_{s,k+j}. \quad (4)$$

This compact representation preserves all essential visual information while exposing only the dynamic patches, enabling tokenization complexity to grow sub-linearly with the number of frames.

### 3.2.2 MOTION-COMPENSATED GATED TOKENIZER

To efficiently tokenize only the dynamic patches identified by our Pixel-Coding Video Scene Representation, we re-purpose the pretrained ViT tokenizer without additional training. We introduce a gating mechanism that filters out static patches using a binary mask derived from patch-wise structural similarity. Specifically, let $P_{s,j}^{(n)}$ denote the $n$-th patch of frame $f_{s,j}$ in scene $s$, and let $\text{SSIM}(\cdot, \cdot)$ be the structural similarity index. We compute the mask entries $M_{s,j}^{(n)}$ as:

$$M_{s,j}^{(n)} = \begin{cases} 1 & \text{if } \text{SSIM}\left(P_{s,j}^{(n)}, P_{s,j-1}^{(n)}\right) < \tau, \\ 0 & \text{otherwise.} \end{cases} \quad (5)$$

where $\tau$ is a fixed threshold. We always set $M_{s,k}^{(n)} = 1$ for the key frame $f_{s,k}$. Next, the gating vector for frame $f_{s,j}$ is formed as

$$G_{s,j} = \left[ M_{s,j}^{(1)}, M_{s,j}^{(2)}, \ldots, M_{s,j}^{(N)} \right], \quad (6)$$

where $N$ is the total number of patches per frame. The gated tokenizer applies $G_{s,j}$ to the ViT embedding layer, processing only the selected patches and inserting zero-vectors for masked positions prior to positional encoding. This design reduces tokenization complexity to sub-linear in the number of frames while preserving all informative content.

### 3.2.3 TOKENIZER ARCHITECTURE

The standard video tokenizer comprises a convolutional patch embedding layer, positional embeddings, and a 26-layer Transformer encoder. Because the embedding convolution uses non-overlapping kernels (stride = kernel size), we can flatten it into an equivalent MLP. Let $p_n$ be the $n$th patch in the gated sequence and $M_n \in \{0,1\}$ its gate mask. We first compute the patch embedding:

$$e_n = \begin{cases} W_c p_n + b_c, & M_n = 1, \\ \mathbf{0}, & M_n = 0, \end{cases} \quad (7)$$

where $W_c$ and $b_c$ are the convolutional kernel weights and bias.

Next, we insert zero placeholders for masked positions and apply positional encoding:

$$\tilde{e}_n = \text{PE}(e_n). \quad (8)$$

Finally, the sequence $\{\tilde{e}_n\}_{n=1}^N$ is processed by the Transformer encoder:

$$v = \text{Transformer}\big(\tilde{e}_1, \tilde{e}_2, \ldots, \tilde{e}_N\big). \tag{9}$$

By gating patches before the embedding layer and using placeholders during positional encoding, we remove static patches while preserving token positions, achieving sub-linear growth in computation with respect to frame count.

### 3.3 SEMANTIC-CODING TOKEN-SCENE MERGING

#### 3.3.1 SEMANTIC-SCENE REPRESENTATION

Pixel-level residual coding captures only low-level changes and lacks semantic context, which is insufficient for high-FPS video understanding. Incorporating deep semantic features often incurs significant computational overhead. However, pretrained vision-tokenizers already embed rich semantic information into discrete tokens. We therefore leverage these existing tokens to perform scene-level merging without retraining.

Let $\mathcal{T}_{s,k}$ be the set of tokens extracted from the key frame of scene $s$, and $\mathcal{T}_{s,k+j}$ the set of tokens from the $j$-th P frame. The full token sequence for scene $s$ up to frame $k+i$ is:

$$\mathcal{T}_{s,k+i} = \mathcal{T}_{s,k} \parallel \mathcal{T}_{s,k+1} \parallel \mathcal{T}_{s,k+2} \parallel \cdots \parallel \mathcal{T}_{s,k+i}. \tag{10}$$

where $\parallel$ denotes sequence concatenation.

To merge semantically similar scenes, we compute a distance between their key-token distributions. For example, using cosine distance:

$$d\big(\mathcal{T}_{s,k}, \mathcal{T}_{t,k}\big) = 1 - \frac{\langle \mu(\mathcal{T}_{s,k}), \, \mu(\mathcal{T}_{t,k}) \rangle}{\|\mu(\mathcal{T}_{s,k})\| \, \|\mu(\mathcal{T}_{t,k})\|}, \tag{11}$$

where $\mu(\cdot)$ computes the mean embedding of a token set. If $d(\mathcal{T}_{s,k}, \mathcal{T}_{t,k}) < \delta$, we merge scene $t$ into $s$ by concatenating its P-frame tokens onto $\mathcal{T}_{s,k}$. This preserves dynamic information while eliminating redundant key-frame tokens.

By merging semantically similar scenes, we further compress the token sequence without losing critical static or dynamic content, enabling efficient high-FPS video processing.

#### 3.3.2 TOKEN-SCENE MERGING

After gated tokenization, each scene $s$ is represented by a *key-token set* $\mathcal{T}_{s,k}$ (from the K-frame) and a sequence of *P-token sets* $\{\mathcal{T}_{s,k+1}, \ldots, \mathcal{T}_{s,k+i}\}$. Since key-token sets dominate the total token count, we merge semantically similar scenes based on the Jensen–Shannon divergence between their normalized token distributions:

$$D_{\text{JSD}}(s,t) = \text{JSD}\big(P(\mathcal{T}_{s,k}), \, P(\mathcal{T}_{t,k})\big), \tag{12}$$

where $P(\mathcal{T})$ denotes the frequency distribution of tokens in $\mathcal{T}$. Given a threshold $\delta$, if

$$D_{\text{JSD}}(s,t) < \delta, \tag{13}$$

we merge scene $t$ into scene $s$. The merged key-token set $\mathcal{T}'_{s,k}$ is formed by averaging the mean embeddings of both sets:

$$\mathcal{T}'_{s,k} = \frac{1}{2}\Big(\mu(\mathcal{T}_{s,k}) + \mu(\mathcal{T}_{t,k})\Big), \tag{14}$$

where $\mu(\mathcal{T})$ computes the mean token embedding of $\mathcal{T}$. Finally, we concatenate the P-token sequences from both scenes:

$$\{\mathcal{T}_{s,k+1}, \ldots, \mathcal{T}_{s,k+i}, \, \mathcal{T}_{t,k+1}, \ldots, \mathcal{T}_{t,k+j}\}. \tag{15}$$

This merging preserves dynamic information while eliminating redundant static tokens, further reducing the overall token sequence length.

## 3.4 IMPLEMENTATION DETAILS

After token-scene merging, we flatten the token sets into a single sequence and pass them through a linear vision-projection layer. These visual tokens are then concatenated with the query prompt and any additional text tokens before being fed, without further training, into the video LLM to generate the answer. This architecture enhances the model's ability to process dense temporal information while remaining compatible with standard LLM pipelines.

## 4 DENSE INFORMATION VIDEO EVALUATION

### 4.1 BENCHMARK CONSTRUCTION

While we have defined the Dense Video Understanding task and proposed a corresponding model, a new benchmark is still required to evaluate performance on this setting. Constructing a question–answer dataset that enforces genuine frame-by-frame reasoning demands preservation of dense temporal information across long video segments.

We introduce the Dense Information Video Evaluation (DIVE) benchmark, designed to stress-test the temporal reasoning and token throughput of video LLMs under high-frame-rate conditions. Unlike prior benchmarks that rely on sparse sampling, DIVE leverages existing video datasets and their subtitle streams to retain fine-grained temporal fidelity. The construction proceeds in two stages:

1. **Video Selection.** We source videos from YouTube (e.g., the LPMDataset (Lee et al., 2023)) with durations of at least 30 minutes. These long-form clips contain over $10^5$ frames at standard frame rates, forcing models to maintain a high sampling rate to capture all content.

2. **Subtitle-Based QA Generation.** Instead of manually authoring frame-specific questions, we exploit the embedded subtitles as ground-truth answers. Subtitles typically update at sub-second intervals; reconstructing the full subtitle stream requires processing tens of thousands of frames.

This design ensures that any model achieving high accuracy must (i) process video at an effectively high FPS and (ii) handle long token sequences without dropping key information. Models that sample too sparsely will either hallucinate subtitle text or omit segments entirely.

### 4.2 ANNOTATION DETAILS

Our annotation pipeline builds on existing YouTube lectures. We first extract time-stamped subtitle files using the Google Subtitle Annotator. These subtitles are then hard-coded into the video frames via FFmpeg's burn-in filter (Tomar, 2006). Finally, we align each subtitle segment with its corresponding frame range to generate QA pairs of the form: **Question:** "What is the subtitle in this video segment?" **Answer:** [Full subtitle text]. By automating QA generation through precise subtitle alignment, we focus evaluation squarely on a model's ability to preserve dense temporal cues and to process lengthy token sequences without information loss.

## 5 EXPERIMENTS

### 5.1 IMPLEMENTATION DETAILS

All experiments are performed in a zero–shot, inference-only setting using both 0.5B- and 7B-parameter variants of our model, built on the LLaVA-One Vision framework (Li et al., 2024) and the QWen2 architecture (Wang et al., 2024). We tokenize 224×224 frames into 16×16 patches using `QwenTokenizerFast`. Inference runs on NVIDIA RTX 4090, RTX A6000 Ada, and H200 GPUs with mixed-precision (FP16). We measure *tokenization time* from raw frame extraction through completion of the token sequence using the LMMS-Eval toolkit (Zhang et al., 2024a). Subjective quality is assessed via Mean Opinion Scores (MOS) assigned by GPT-3.5 (Singh & Singh, 2023), where each generated answer is rated against the ground truth on a 0–5 scale.

| Method | #Parameters | MOS |
|---|---|---|
| LLaVA-Video (Zhang et al., 2024b) | 7B | 1.47 |
| LLaVA-OneVision (Li et al., 2025) | 7B | 1.70 |
| LLaVA-OneVision (Li et al., 2025) | 0.5B | 2.01 |
| LLaVA-Next SI (Li et al., 2024) | 0.5B | 1.73 |
| **GRT (Ours)** | 0.5B | **2.50** |

Table 1: Mean Opinion Scores (MOS) of different methods on the DIVE benchmark. Despite using only a 0.5B-parameter backbone, our method outperforms several larger 7B and comparable 0.5B video-LLM baselines.

| Method | 0.01 FPS | 0.1 FPS | 1 FPS |
|---|---|---|---|
| LLaVA-OV (Li et al., 2025) | 0.0170 s | 0.0186 s | 0.0487 s |
| GRT (Ours) | 0.0174 s | 0.0177 s | 0.0226 s |
| *Speedup* | 10.2% | 5.1% | 46.4% |

Table 2: Tokenization time across fixed frame rates (FPS). Consistent speedups at various FPS.

| | MLVU (Zhou et al., 2025) | VideoMME (Fu et al., 2025) |
|---|---|---|
| Baseline (Li et al., 2025) | 33.002 | 43.963 |
| Our Method | 34.066 | 44.037 |
| Δ (Our − Base) | +1.064 | +0.074 |

Table 3: Performance on existing benchmarks.

## 5.2 COMPARATIVE METHODS

As our work is the first to target the task of dense video understanding, there is no existing literature directly comparable to ours. To ensure a fair evaluation, we adapt several recent video large language models (video-LLMs) as baselines by modifying their backbones to support denser frame processing. We compare our approach against five recent video-LLM systems. 1) LLaVA-One Vision (Li et al., 2025) samples a fixed number of frames at uniform intervals before concatenating visual and text tokens. Its flexible FPS variant (Kim et al., 2024) dynamically adjusts the sampling rate to focus on high-motion segments. 2) LLaVA-Video (Zhang et al., 2024b) integrates temporal-attention modules into the LLaVA backbone to capture inter-frame motion cues. 3) LLaVA-Next SI (Li et al., 2024) introduces spatial-instruction tuning and temporal-aware modifications into LLaVA with single image finetuning to improve instructional video understanding.

## 5.3 EVALUATION METRICS

We evaluate all methods using a combination of objective and subjective metrics. Mean Opinion Score (MOS) (Streijl et al., 2016) reflects the perceived answer quality on a 0–5 scale, as judged by GPT-3.5. Tokenization time measures the wall-clock duration required to convert raw video frames into visual tokens. Accuracy is defined as the binary correctness of each QA pair; however, due to the open-ended nature of our dense-FPS questions, we emphasize MOS in our primary comparisons. Effective FPS quantifies the average number of frames processed per second during inference.

## 5.4 QUANTITATIVE RESULTS

Table 1 presents the Mean Opinion Scores (MOS) on the DIVE benchmark. Our 0.5B-parameter model achieves an MOS of 2.50, outperforming all baselines—including the larger 7B-parameter LLaVA-Video (1.47) and both 0.5B and 7B variants of LLaVA-OV and LLaVA-SI. These results demonstrate that our gated tokenization and semantic-scene merging strategies yield substantial gains in answer quality without increasing model size.

## 5.5 IMPACT OF FRAME RATE

To assess robustness to varying temporal resolutions, we vary the input frame rate from 0.0001 FPS to 1.0 FPS. For a controlled comparison, we remove the maximum-frame constraint from the LLaVA-OV baseline, ensuring both methods share the same model size (0.5B). Figure 6 plots MOS as a function of FPS. Our model's performance increases steadily with higher FPS, indicating effective utilization of dense temporal cues. In contrast, the modified baseline exhibits a sharp performance decline at very low FPS, with only modest recovery at higher rates. Across all tested frame rates, our method maintains a clear lead, underscoring the need for high-FPS evaluation in vLLMs.

## 5.6 ABLATION STUDY

To quantify the contributions of our two main components, the Motion-Compensated Gated Inter-Tokenizer ("Gated Tokenizer") and the Semantic-Scene Merging module ("Scene Merge"), we perform an ablation study on the DIVE benchmark. Table 4 reports both accuracy and MOS for three configurations: **Baseline (no components)**: raw full-FPS tokenization without any gating or merging. **Gated Tokenizer only**: applies pixel-coding gating to prune static patches before tokenization. **Full**

Table 4: Ablation results showing the impact of each module. "—" indicates disabled, and "✓" indicates enabled.

| Gated Tokenizer | Scene Merge | Accuracy | MOS |
|---|---|---|---|
| — | — | 0.1152 | 1.66 |
| ✓ | — | **0.1451** | 1.93 |
| ✓ | ✓ | 0.1262 | **1.94** |

Table 5: Effect of frame rate on token retention after each reduction stage. Values represent the proportion of tokens retained relative to baseline.

| Reduction Stage | 0.01 FPS | 0.1 FPS | 1 FPS |
|---|---|---|---|
| Gated Pruning | 1.00 | 0.96 | 0.90 |
| Scene Merging | 1.00 | 0.33 | 0.14 |

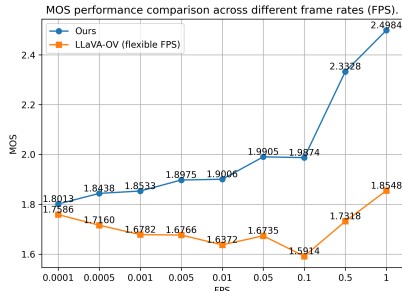

Table 6: MOS versus input frame rate (FPS) for our model and the modified LLaVA-OV baseline. Higher FPS leads to improved performance, and our method consistently outperforms baseline.

**model**: combines the Gated Tokenizer with Semantic-Scene Merging. Enabling the Gated Tokenizer alone yields a significant boost in both accuracy and MOS compared to the baseline, demonstrating the effectiveness of early patch pruning. Incorporating the Scene Merge module further increases MOS—indicating richer, more informative answers—while incurring a small drop in accuracy due to the higher compression rate. Overall, the full model provides the best balance of efficiency and answer fidelity, demonstrating that both components are critical for optimal high-FPS video understanding.

### 5.7 FURTHER EVALUATION ON EXISTING BENCHMARKS

To demonstrate generalization beyond DIVE, we evaluated GRT on two established long-form video benchmarks: MLVU (Zhou et al., 2025) and VideoMME (Fu et al., 2025). We applied our method with the same sampling rate as the LLaVA-One Vision (0.5B) baseline (Li et al., 2024). As Table 3 shows, GRT achieves consistent improvements, particularly on MLVU, while using fewer tokens and reducing tokenization time. These results confirm the applicability of our approach across both dense high-FPS and traditional video-understanding settings.

### 5.8 TOKENIZATION TIME AND TOKEN REDUCTION UNDER DIFFERENT FPS

We evaluate both tokenization latency and token reduction effectiveness at three representative frame rates: 0.01, 0.1, and 1 FPS. Table 2 reports the average time to tokenize a short video segment for our method versus the LLaVA-OV baseline (which processes every patch without gating or merging). Timing is measured by instrumenting the tokenizer entry and exit points on an NVIDIA RTX 4090. At 1 FPS, our motion-compensated gated tokenizer reduces latency by 46.4%, from 0.0487 s to 0.0226 s. Even at lower frame rates, we observe improvements of 5.1% and 10.2% at 0.1 FPS and 0.01 FPS, respectively, demonstrating that early patch pruning yields consistent speed-ups as the number of frames grows.

To quantify token savings, Table 5 shows the fraction of original tokens retained after each reduction stage. At 0.01 FPS, both gated pruning and scene merging have little effect, since few inter-frame changes occur. At 0.1 FPS, gated pruning alone retains 96% of tokens, while combining with scene merging reduces this to 33%. At 1 FPS, gated pruning retains 90% of tokens, and scene merging further reduces the sequence to just 14%. These results confirm that our two-stage reduction, first at the patch level, then at the scene level, becomes increasingly effective as FPS increases.

### 6 CONCLUSION

We introduced the task of **Dense Video Understanding**, which requires video LLMs to reason over densely sampled, high-FPS content. To support this, we proposed **DIVE**, the first benchmark for evaluating fine-grained temporal understanding in video QA. To address the inefficiency of high-frame-rate tokenization, we developed *Gated Residual Tokenization (GRT)*, a two-stage framework that combines motion-compensated gating and semantic-scene merging to reduce token count with minimal information loss. Experiments on 0.5B and 7B models show that GRT achieves state-of-the-art accuracy and MOS, while accelerating tokenization by up to 46%. Ablation studies confirm the effectiveness of both gating and merging, highlighting the value of dense temporal modeling for video large language models.

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

# A   APPENDIX

## A.1   ETHICS STATEMENT

Our study adheres to ethical research practices. All video data used in this work are from public datasets. No personally identifiable information (PII) or sensitive data is included in the dataset. We are committed to ensuring that the benchmark and models released upon acceptance are used solely for research and educational purposes, and we will include explicit licensing terms to prevent misuse in areas such as surveillance or privacy-intrusive applications.

## A.2   REPRODUCIBILITY STATEMENT

**Under review (double-anonymous):** We will release the full benchmark, evaluation code, and model scripts under a permissive research license, and contribute our task adapters to the `lmms-eval` evaluation toolkit via a public pull request. A reproducibility README will enumerate environment setup, seeds, and commands for end-to-end replication.

## A.3   LARGE LANGUAGE MODEL USAGE DETAILS

**Writing assistance.** The manuscript was conceived and written by the authors. Large language models (LLMs) were used *only* for light copy-editing (grammar, phrasing) and consistency checks. No model generated technical content, claims, experimental designs, or results.

**Data annotation support.** For dataset preparation, we used a generic, browser-based subtitle annotation/alignment tool to accelerate timestamp correction and segment labeling. Annotations were verified by human annotators; automated assistance did not make final labeling decisions.

## A.4   LIMITATIONS

While our method demonstrates strong performance on short to medium-length videos, its effectiveness diminishes as video length increases. This is because our merging strategy heavily relies on strong temporal redundancy, which becomes sparse in longer videos. Additionally, the sparse sampling over extended sequences may lead to missed key content, making the overall representation less reliable.

Although we propose the first-ever benchmark for dense video understanding, a major challenge remains in devising diverse question–answer pairs that truly capture the full spectrum of dense, frame-by-frame information. Automatically generating or manually annotating such tasks, where every frame's content contributes meaningfully, remains non-trivial. We regard this as an important future direction to explore.

## A.5   REAL-WORLD APPLICATIONS AND FUTURE DIRECTIONS

Dense video understanding is critical for any application requiring frame-by-frame analysis of rapidly changing visual content. Unfortunately, no publicly available video-LLM benchmarks currently capture this need—most datasets sample at low frame rates and do not stress-test models on dense temporal information. We designed DIVE to fill this gap for educational lecture videos, but many other domains urgently require similar resources.

Sign language translation involves rapid hand and finger movements; one study found that intelligibility of mobile sign language video declines sharply when transmitted below recommended frame rates (e.g., from 30 FPS down to 15 FPS) (Tran et al., 2014). Surgical video analysis—such as endoscopy—also demands high temporal resolution: below 15 FPS, clinicians risk missing critical procedural details, and specialized high-speed modalities can exceed 100 FPS to capture tissue dynamics (Wei et al., 2025). Sports analytics benefits from 30–60 FPS sampling to track split-second actions without motion artifacts (Banitalebi-Dehkordi et al., 2014). In surveillance systems, detection performance degrades significantly when frame rates fall below 8 FPS (Keval & Sasse, 2008). Even consumer applications—such as high-FPS gaming and live streaming—depend on dense sampling for smooth, interactive experiences.

| Domain | Min. FPS Needed | Recommended FPS | Source |
|---|---|---|---|
| Sign language | 15 FPS | 30 FPS or higher | (Tran et al., 2014) |
| Surgical (endoscopic) | 15 FPS | 30 FPS | (Wei et al., 2025) |
| Sports analytics | 24 FPS | 30–60 FPS | (Banitalebi-Dehkordi et al., 2014) |
| Surveillance (CCTV) | 8 FPS | 8–15 FPS | (Keval & Sasse, 2008) |
| High-FPS gaming | 60 FPS | 120 FPS or higher | Industry practice |

Table 7: Frame-rate requirements across diverse application domains.

Table 7 shows that many real-world scenarios demand higher frame rates than current video-LLM benchmarks provide. We envision future extensions of DIVE—such as *DIVE-Sign*, *DIVE-Surg*, and *DIVE-Sport*—to cover these domains, enabling comprehensive evaluation of video LLMs on dense temporal reasoning tasks. Creating and releasing such datasets is a promising direction to ensure that next-generation video LLMs meet the needs of safety-critical and high-precision applications.

