# OpenReview forum: "Dense Video Understanding with Gated Residual Tokenization"
_ICLR.cc/2026/Conference — ICLR 2026 Conference Withdrawn Submission_

### Official Review · Reviewer_DJ9m · 2025-11-01

**Soundness:** 3
**Presentation:** 4
**Contribution:** 3
**Rating:** 4
**Confidence:** 5

**Summary:**

This paper defines the task of "dense video understanding" and provides the first evaluation benchmark DIVE and an efficient solution GRT. GRT enables VLLM to efficiently process high-frame-rate videos through a two-stage gated and merging strategy, validating the importance of dense temporal information for video understanding.

**Strengths:**

1. For the first time proposed the task of "Dense Video Understanding" (DVE) and built a new benchmark (DIVE): by using subtitles as answer ground truth, it forces the model to perform high-frame-rate frame-by-frame reasoning, which better reflects the model's ability to process dense information compared to traditional benchmarks.
2. GRT framework is a well-designed two-stage system. It first filters static content at the pixel level through "gated tokenization," and then compresses similar scenes at the semantic level through "scene merging."

**Weaknesses:**

1. Baseline Diversity (DIVE): The authors also mention this in the limitations section (A.4). The current DIVE baseline heavily relies on subtitles (OCR) as a task to measure "dense information." This leads to a limited evaluation of the "dense understanding" dimension. Therefore, a model may perform well on DIVE (because it is good at OCR), but perform poorly on tasks that truly require understanding dense motion or interactions.
2. Generalization ability is limited on traditional benchmarks: when the GRT method is applied to existing long video benchmarks (such as MLVU and VideoMME), its performance improvement is very limited. This may indicate that GRT is a highly specialized optimization method suitable for its defined "dense tasks," but its advantages are not significant for traditional, content-slow-changing long video tasks.
3. The second stage of GRT, called "Scene Merge," is a lossy compression, as shown in the ablation study (Table 4) of the paper. When only using "Gated Tokenizer," the model's "accuracy" is 0.1451. But when adding "Scene Merge," the "accuracy" drops to 0.1262.

**Questions:**

1. This method introduces at least two key hyperparameters, $\tau$ (SSIM threshold) and $\delta$ (JSD threshold), which are not discussed in detail in the paper.

---

### Official Review · Reviewer_Cbno · 2025-11-02

**Soundness:** 1
**Presentation:** 2
**Contribution:** 1
**Rating:** 2
**Confidence:** 4

**Summary:**

In this paper, a new dataset, called DIVE, is proposed for dense video understanding. This dataset consists of YouTube videos and it is annotated for substitle prediction task. To achieve high accuracy, it requires more frames than existing benchmark tests. Also, the gated residual tokenization is proposed to handle large number of input frames more efficiently. Specifically, it selects the patches based on motion scale and also merges the tokens based on Jensen-Shannon divergence. Experimental results show that the proposed algorithm achieves better MOS score than existing methods on the DIVE dataset.

**Strengths:**

- This paper is clearly written.
- This paper proposes a new benchmark dataset, designed to stress-test the video LLMs.

**Weaknesses:**

- Using only MOS as the evaluation metric does not seem sufficient. Since the task is subtitle prediction, it should be possible to use accuracy-based metrics. The rationale for not using such metrics is not clearly explained beyond Lines 405–407, and further clarification would strengthen the paper. Additionally, alternative metrics such as BLEU could be considered instead of a purely binary evaluation.
- Reporting how accuracy evolves with increasing frame counts would help verify the authors’ claim that the dataset is densely informative in the temporal dimension.
- The dataset description is not sufficiently detailed. It would be beneficial to include statistics such as the total number of videos, their average duration and frame rate (FPS), the number and average length of subtitles, and the average time interval between subtitle changes. Providing example instances from the dataset would also enhance clarity.
- Given that the 0.5B model significantly outperforms the 7B model in Table 1, it is unclear whether this dataset can be regarded as sufficiently challenging.
- It would be helpful to include a comparison of inference examples.
- The proposed method performs extensive token reduction, yet it achieves better performance than other models. It would be necessary to provide an analysis showing how well the subtitle-related regions or information are preserved after token reduction.

**Questions:**

Please see my concerns in the weakness section.

---

### Official Review · Reviewer_ZHhD · 2025-11-03

**Soundness:** 2
**Presentation:** 3
**Contribution:** 2
**Rating:** 2
**Confidence:** 4

**Summary:**

This paper studies a critical limitation in existing video large language models: low-frame-rate sampling. The authors argue this trade-off, made to avoid prohibitive computational costs, prevents models from understanding fine-grained temporal details.

To address this, the paper makes three core contributions:

1. Task Definition: It introduces "Dense Video Understanding," a task challenging models to process high-frame-rate (high-FPS) video.
2. New Benchmark (DIVE): The benchmark which focuses on high FPS.
3. Novel Method (GRT): It presents Gated Residual Tokenization (GRT), a two-stage token reduction framework designed for efficiency in high-FPS video:
    * Stage 1: Motion-Compensated Gated Inter-Tokenization: Inspired by video compression (K-frames and P-frames), this stage uses a motion mask (via SSIM) to identify static vs. moving patches. It only runs the expensive tokenization process on the "moving" patches (P-frames), achieving sub-linear growth in token count as FPS increases.
    * Stage 2: Semantic-Scene Token Merging: This stage further compresses the token sequence by identifying and merging entire scenes that are semantically similar (based on their key-frame token distributions).


In addition, the authors’ 0.5B model performs significantly better than the chosen 7B baseline model.

**Strengths:**

1. Addresses an Important and Under-Studied Problem: The paper correctly identifies a key limitation of current VLLMs—their reliance on sparse sampling—and tackles the important and challenging problem of enabling efficient, high-FPS video understanding.
2. New Two-Stage Method: The proposed Gated Residual Tokenization (GRT) is a new solution that operates at two levels: it uses low-level motion information to prune redundant patches before tokenization and high-level semantic information to merge redundant scenes after tokenization.
3. Clarity and Presentation: The paper is well-written, clear, and easy to follow. The core concepts and the motivation are communicated effectively.

**Weaknesses:**

1. Insufficient Comparison to Related Work: The paper fails to compare GRT against several highly relevant existing works. The "Motion-Compensated Gated Inter-Tokenization" appears very similar to methods [1, 2] that also skip patches based on similarity, and the motion/Key frame/P frame-based approach is related to [3] which models motion vectors. A direct comparison is essential to validate the novelty and superiority of GRT.
2. Lack of Experimental Clarity and Ablation: The paper does not clearly explain the experimental setup for choosing K-frames and P-frames, especially in the 1 FPS setting. Furthermore, no ablation study is provided on the impact of key-frame frequency, which is a critical hyperparameter of the proposed method.
3. Misalignment of DIVE Benchmark with Motivation: The DIVE benchmark, while novel, seems misaligned with the paper's core motivation. The task is effectively dense Optical Character Recognition (OCR), not general dense temporal reasoning. Subtitle information can often be more efficiently extracted as text or audio streams, which undermines the argument that visual frame-by-frame processing is necessary. This OCR-heavy task does not guarantee the method will generalize to complex action or motion patterns (e.g., sign language, sports) which usually requires high FPS as discussed in appendix.
4. Evaluation Does Not Use "High FPS": A major flaw is that the experiments are conducted at low frame rates (0.0001 to 1.0 FPS). This can hardly be considered "high FPS." Many existing VLLMs (e.g., Gemini, Qwen) can already process long videos at 1 FPS. To substantiate its claims, the paper should include experiments at genuinely high frame rates (e.g., >4 FPS, 15 FPS, or 30 FPS).
5. Missing Scalability Results: The paper demonstrates GRT's effectiveness on a 0.5B model but does not show results for larger models (e.g., 7B) e.g. in the Table1. It is important to demonstrate that the method's benefits scale.
6. Unclear Mechanism for DIVE Performance: Given that the DIVE benchmark relies on subtitles (which are small, static-text regions) and frames are resized to 224x224, it is plausible that the motion-gating mechanism might identify subtitle patches as "static" and prune them. The paper provides no analysis to confirm that these critical, information-bearing patches are being preserved. Without this, it is unclear why the model is performing well.
7. Marginal Gains on Standard Benchmarks: The improvements on existing benchmarks (MLVU, VideoMME) are marginal. This suggests the method's benefits may be highly specific to the new DIVE benchmark. Evaluating on a other existing action-focused benchmark (e.g., ActionAtlas [4]) would be beneficial.


[1] Don’t Look Twice: Faster Video Transformers with Run-Length Tokenization https://arxiv.org/pdf/2411.05222

[2] ELASTICTOK: ADAPTIVE TOKENIZATION FOR IMAGE AND VIDEO https://arxiv.org/pdf/2410.08368

[3] Video-LaVIT: Unified Video-Language Pre-training with Decoupled  Visual-Motional Tokenization https://arxiv.org/pdf/2402.03161

[4] ActionAtlas: https://arxiv.org/pdf/2402.03161

**Questions:**

* Can you provide a detailed comparison against methods [1, 2] that also remove patches based on similarity, and [3] which models motion vectors? How does GRT's performance and computational overhead differ?
* What was the specific heuristic or algorithm used to determine key-frame (K-frame) placement in your experiments? Have you conducted an ablation study on the frequency of K-frames, and how does this hyperparameter affect the model's performance and efficiency?
* Since the DIVE benchmark is effectively a dense OCR task, how can you be sure the model is learning general-purpose dense temporal reasoning rather than just becoming a better frame-by-frame text recognizer? Why is processing these frames visually superior to extracting the subtitle information from text or audio streams, which would be far more efficient?
* The central claim is about "high-FPS" understanding, yet the evaluation only extends to 1.0 FPS. Can you provide results on high frame rates (e.g., > 4 FPS) to demonstrate that your method's benefits hold and scale in the scenarios you motivate?
* Your primary results are on a 0.5B parameter model. How does Gated Residual Tokenization (GRT) perform when integrated into much larger, state-of-the-art models (e.g., 7B or higher)? Do the efficiency and accuracy benefits scale?
* Given the small 224x224 resolution, it's plausible your motion-gating mechanism (SSIM) might incorrectly identify static subtitle text as "unchanged" and prune these critical patches. Have you analyzed which patches are being pruned vs. kept? Can you confirm that the subtitle patches are consistently being processed?
* The performance gains on standard benchmarks (MLVU, VideoMME) are marginal, suggesting GRT might be highly tuned for the DIVE (OCR) task. How does the method perform on benchmarks that are highly focused on fine-grained action and motion (e.g., ActionAtlas, or sign language datasets)?

[1] Don’t Look Twice: Faster Video Transformers with Run-Length Tokenization https://arxiv.org/pdf/2411.05222

[2] ELASTICTOK: ADAPTIVE TOKENIZATION FOR IMAGE AND VIDEO https://arxiv.org/pdf/2410.08368

[3] Video-LaVIT: Unified Video-Language Pre-training with Decoupled  Visual-Motional Tokenization https://arxiv.org/pdf/2402.03161

[4] ActionAtlas: https://arxiv.org/pdf/2402.0316

---

### Official Review · Reviewer_Ck3k · 2025-11-04

**Soundness:** 2
**Presentation:** 3
**Contribution:** 2
**Rating:** 4
**Confidence:** 3

**Summary:**

This paper introduces the new task of Dense Video Understanding, aiming to enable video large language models to process and reason over high fps videos that retain fine-grained temporal information. Current VLLMs often sample only a few frames to reduce computational cost, discarding dense temporal cues essential for tasks such as lecture or instructional video comprehension. To overcome this limitation, the authors propose Gated Residual Tokenization, a two-stage framework combining motion-compensated inter-tokenization and semantic-scene token merging to skip static regions and merge redundant tokens, achieving sub-linear token growth and reduced tokenization time. They further introduce DIVE, the first benchmark designed to evaluate high fps video QA with frame-by-frame reasoning. Experiments show that GRT significantly improves efficiency and answer quality over baselines.

**Strengths:**

1.The paper introduces the new concept of Dense Video Understanding, highlighting the overlooked need for fine-grained, high fps reasoning in video LLMs.
2.The proposed benchmark fills an important evaluation gap by enabling quantitative assessment of dense temporal reasoning capabilities, beyond conventional sparse-sampling datasets.
3.The Gated Residual Tokenization framework cleverly integrates motion compensated gating and semantic scene merging, achieving sublinear tokenization cost without sacrificing temporal fidelity.

**Weaknesses:**

1.The experiments focus mainly on the proposed DIVE benchmark, which is constructed from lecture and subtitle videos. Broader validation on diverse high-FPS domains (e.g., sports, egocentric, or dynamic scenes) is missing.
2.There is no qualitative or visualization analysis showing how GRT modifies token distributions or captures motion semantics compared to standard tokenizers, leaving the interpretability of the method unclear.
3.The ablation analysis is minimal, only on two components, without examining the other components of the proposed method.

**Questions:**

see weakness

---

### Note · Authors · 2025-11-14

**Comment:**

Dear Reviewers and ACs,

We sincerely appreciate your time, effort, and valuable suggestions in reviewing our submission. Although we decided not to submit a full rebuttal due to time constraints and will withdraw the paper from further consideration, we would still like to briefly clarify our perspective and share our planned revisions. We also feel that there are a few misunderstandings about the scope and novelty of our work, which we will clarify and address in future versions.

First, regarding the main contribution: our primary focus is to introduce the Dense Video Understanding task as, to our knowledge, the first benchmark that explicitly emphasizes the importance of dense frame sampling for Video LLMs. There is no inherent connection between OCR and dense video understanding; OCR per frame is simply one possible mechanism tool to build an evaluation interface that checks whether a VLM model can correctly reason over information present in every frame within a single QA instance.

Second, concerning gated tokenization: our intent is to propose, to our knowledge, the first approach that prunes tokens before they are generated by the tokenizer, thereby saving both tokenization time and computation. Its generalizability is supported by the widespread adoption of H.264 video coding. Pruning inside the tokenization process is fundamentally different from existing token pruning and token merging methods, which operate after tokens have already been produced.

Although we are not rebutting in detail this round, we have carefully read your comments and plan to improve the work along the following directions:

1. Motion-dense subtask: We will add a motion-focused dense subtask as a complementary component to the benchmark, explicitly examining motion information across all frames, which is especially important for dense video understanding.

2. Clarifying task scope: We will further clarify why our work is not simply an OCR task and not just another long-video understanding benchmark, but instead targets dense, frame-by-frame reasoning.

3. More precise definition of “dense”: We will adopt a more specific and principled way to define dense video frames, beyond just using frame rate, to better capture the core properties of dense temporal information.

We will carefully clarify these misunderstandings and revise the paper accordingly in future versions. Even though we chose not to submit a rebuttal this time, we are truly grateful for your efforts and constructive feedback. Your comments will significantly help us strengthen the Dense Video Understanding task and benchmark in our future revisions.

Best regards,
The Authors

**Withdrawal Confirmation:**

I have read and agree with the venue's withdrawal policy on behalf of myself and my co-authors.